# Transient Effect of Infant Formula Supplementation on the Intestinal Microbiota

**DOI:** 10.3390/nu13030807

**Published:** 2021-03-01

**Authors:** Ning Chin, Gema Méndez-Lagares, Diana H. Taft, Victoria Laleau, Hung Kieu, Nicole R. Narayan, Susan B. Roberts, David A. Mills, Dennis J. Hartigan-O’Connor, Valerie J. Flaherman

**Affiliations:** 1California National Primate Research Center, University of California, Davis, CA 95616, USA; nchin@ucdavis.edu (N.C.); gemendez@ucdavis.edu (G.M.-L.); htkieu@ucdavis.edu (H.K.); nicolenarayan@gmail.com (N.R.N.); 2Department of Medical Microbiology and Immunology, University of California, Davis, CA 95616, USA; 3Department of Food Science and Technology, University of California, Davis, CA 95616, USA; dhtaft@ucdavis.edu (D.H.T.); damills@ucdavis.edu (D.A.M.); 4Foods for Health Institute, University of California, Davis, CA 95616, USA; 5Department of Pediatrics, University of California, San Francisco, CA 94143, USA; victoria.laleau@ucsf.edu (V.L.); valerie.flaherman@ucsf.edu (V.J.F.); 6Friedman School of Nutrition Science and Policy, Tufts University, Boston, MA 02111, USA; susan.roberts@tufts.edu; 7Department of Viticulture and Enology, University of California, Davis, CA 95616, USA; 8Division of Experimental Medicine, Department of Medicine, University of California, San Francisco, CA 94143, USA; 9Department of Epidemiology and Biostatistics, University of California, San Francisco, CA 94143, USA

**Keywords:** infant diet, microbiome, immune development, formula supplementation, breastfeeding, delivery mode, microbiota–immune system interactions

## Abstract

Breastfeeding is the gold standard for feeding infants because of its long-term benefits to health and development, but most infants in the United States are not exclusively breastfed in the first six months. We enrolled 24 infants who were either exclusively breastfed or supplemented with formula by the age of one month. We collected diet information, stool samples for evaluation of microbiotas by 16S rRNA sequencing, and blood samples for assessment of immune development by flow cytometry from birth to 6 months of age. We further typed the *Bifidobacterium* strains in stool samples whose 16S rRNA sequencing showed the presence of Bifidobacteriaceae. Supplementation with formula during breastfeeding transiently changed the composition of the gut microbiome, but the impact dissipated by six months of age. For example, *Bifidobacterium longum*, a bacterial species highly correlated with human milk consumption, was found to be significantly different only at 1 month of age but not at later time points. No immunologic differences were found to be associated with supplementation, including the development of T-cell subsets, B cells, or monocytes. These data suggest that early formula supplementation, given in addition to breast milk, has minimal lasting impact on the gut microbiome or immunity.

## 1. Introduction

Exclusive breastfeeding from birth through the age of six months has long been recommended by the American Academy of Pediatrics to be the gold standard for feeding infants [1]. Breastfeeding dramatically reduces the risk of infectious and allergic diseases [2,3,4,5,6,7], in part because human milk oligosaccharides promote the growth of a beneficial microbiota [8]. Such microbiotas protect the newborn enterally by blocking pathogen adhesion [9,10,11] and systemically by inducing robust development of T helper 17 cells and regulatory T cells that maintain gut integrity and dampen inflammatory responses—possibly in cooperation with key bacterial metabolites such as short-chain fatty acids (SFCA) that maintain mucosal integrity [12,13,14,15]. Perhaps the most unique characteristic of the intestinal microbiota among exclusively breastfed infants, as compared to formula-fed infants, is increased abundance of bifidobacteria. *Bifidobacterium longum* subspecies (subsp.) *longum*, *Bifidobacterium longum* subsp. *infantis*, and *Bifidobacterium breve* have been strongly associated with beneficial health outcomes, including preventing enteric inflammation [16], reducing diarrhea [17], and improving allergy symptoms [18].

While exclusive breastfeeding is encouraged, only 24.9% of infants in the United States were exclusively breastfed through 6 months in 2015 [19]. In circumstances including insufficient milk production, inability to breastfeed, or low birth weight, some infants receive supplemental donor breast milk or formula. Exclusively breastfeeding infants were twice as likely to be readmitted to the hospital due to jaundice [20,21] and dehydration [22]. Intervention with formula supplementation in addition to breastfeeding during early life (before 1-month old) showed a trend toward reduced rehospitalization rate and did not interfere with breastfeeding later in life [23], suggesting that fully feeding infants by employing supplementation with formula is more beneficial to infants than exclusively breastfeeding. Additionally, breastfeeding habits vary among different cultures and socioeconomic strata. Mothers in high-income countries generally have shorter breastfeeding duration than mothers in low-income countries [6], possibly resulting in lower abundance of *B. longum* subsp. *infantis* among breastfed infants in high-income countries. This disparity has been hypothesized to account at least partly for the increased prevalence of allergic disease in high-income settings [24].

The impact of intestinal microbiota on infant health is thought to be at least partially mediated by the relationship between intestinal microbiota and the developing human immune system. Gnotobiotic animals, colonized by only one bacterial strain, have dysfunctional innate and adaptive immunity; colonization with specific commensals can partially correct the deficiencies [25,26]. In humans, intestinal abundance of bifidobacteria has been associated with increased salivary immunoglobulin A (IgA) [27], CD20+ B cells [28], and IgA- and IgM-secreting cells [29]. A recent study showed a correlation between bifidobacteria abundance in early infancy and CD4+ T-cell responses to immunizations including those with tetanus toxoid and hepatitis B, and a similar correlation for the subspecies *B. longum* subsp. *infantis* [13,30]. However, existing studies documenting associations between intestinal microbiota and immune responses have not had access to detailed data on dietary intake, which might influence both intestinal microbiota and immune responses. We therefore undertook a prospective, longitudinal study called Milk, Intestinal Microbiota, and Immunity (MIMI) to determine the relationship between dietary intake, intestinal microbiota, and immune development.

## 2. Materials and Methods

### 2.1. Subjects and Sample Collection

We enrolled 24 healthy, exclusively breastfeeding, singleton term (≥37 weeks gestation) infants during the birth hospitalization at the University of California San Francisco (UCSF) Medical Center (Table 1). Infants were excluded if mothers were <18 years of age or were not English-speaking or if breastfeeding was not recommended by the clinical team. Infants who had antibiotic exposure postnatally were excluded from this study. We did not specifically query mothers about antenatal vaccine, but it is the standard practice [31]. We obtained informed consent from mothers for their infants. This study was approved by the UCSF Committee on Human Research (Protocol# 14-13484). Dietary intake data were collected at birth; 1 week; and 1, 3, and 6 months of age. Data on receipt of immunizations were collected at 1, 3, and 6 months of age. Stool specimens were collected with Stool Nucleic Acid Collection and Preservation Tubes (Norgen Biotek Corp., Thorold, ON, Canada) at birth; 1 week; and 1, 3, and 6 months. Meconium was collected on the day of birth. Stool samples were stored at −70 °C until DNA extraction. Blood specimens for the assessment of immune function were collected by heel stick at birth and at 1, 3, and 6 months of age into test tubes containing heparin or ethylenediaminetetraacetic acid (EDTA), shipped to University of California, Davis (UCD), immediately at room temperature, received within 24 hours, and processed immediately after sample receipt: plasma were collected after centrifuging the blood at 2500 rpm for 15 min and stored at −70 °C; peripheral blood mononuclear cells (PBMCs) were isolated by centrifugation of blood cells onto lymphocyte separation medium (MP Biomedicals LLC, Irvine, CA, USA), then cryopreserved by rate-controlled freezing in the presence of 10% dimethyl sulfoxide (DMSO).

### 2.2. Bacterial DNA Extraction and Sequencing

We collected stool samples at all five time points (birth, 1 week, 1 month, 3 months, and 6 months) for 15 subjects (5 cesarean-delivered and 10 vaginally delivered); two time points (birth and 1 week) for 1 subject (vaginally delivered); three time points (birth, 1 week, and 1 month) for 4 subjects (2 cesarean-delivered and 2 vaginally delivered); and four time points (birth, 1 month, 3 months, and 6 months) for 4 subjects (1 cesarean-delivered and 3 vaginally delivered). Stool samples were shipped to UCD after collection and stored at −70 °C until DNA extraction using the MoBio PowerSoil kit (Qiagen Inc., Hilden, Germany). Amplicon libraries were generated by amplifying the V4 variable region of 16S rRNA genes, quality checked, pooled, cleaned, and sequenced using a 250 bp paired-end method on an Illumina MiSeq instrument. The quality of sequencing reads was checked using FastQC [32]. Sequences were trimmed and annotated to the species level using the DADA2 package [33] with Greengenes database v13_8 [34] within the R software environment [35]. The number of reads per sample per bacterial feature was stored in a matrix and used in downstream statistical analysis. Samples found to have sequences from the *Bifidobacteriaceae* family were further subjected to analysis of bifidobacterial-species-specific terminal restriction fragment length polymorphisms (Bif-TRFLP) to identify specific species of *Bifidobacterium* [36].

### 2.3. Immune-Cell Phenotyping by Flow Cytometry

PBMCs were successfully extracted from 65 blood samples and analyzed using flow cytometry. Frequencies of monocytes and T-cell subsets in peripheral blood samples were determined by flow cytometry of cryopreserved PBMCs. The following antibodies were used: anti-CD3-BV786 (clone SP34-2), anti-CD4-PerCP-Cy^TM^5.5 (clone L200), anti-CD8-Alexa 700 (clone RPA-T8), anti-CD95-APC (clone DX2), anti-CCR7-BV421 (clone 3D12), anti-CD127-PE (clone HIL-7R-M21), anti-CD25-PE-Cy^TM^7 (clone M-A251), anti-HLA-DR-ECD (clone Immu-357), anti-CD14-BV605 (clone M5E2), anti-CD16-APC-H7 (clone 3G8), anti-CD20-FITC (clone L27), and anti-CD80-BV650 (clone L307.4). A cell viability dye was included to discriminate live from dead cells (Invitrogen Aqua Live/Dead Fixable Dead Cell Stain, Invitrogen, Carlsbad, CA, USA). Cells were washed and fixed in phosphate-buffered saline containing 1% paraformaldehyde. Data were acquired on LSR II or Fortessa cytometers (BD Biosciences, San Jose, CA, USA) and analyzed using FlowJo^TM^ Software v10.2 (Becton, Dickinson and Company, Ashland, OR, USA).

### 2.4. Vaccine Responses to Tetanus Toxoid

A total of 76 plasma samples were collected: 35 samples before vaccination, 21 samples 8–60 days (median 35 days) post first dose of the HepB-DTaP-IPV-HiB-PCV-RV vaccine (protecting against hepatitis B, diphtheria, tetanus, pertussis, polio, *Haemophilus influenzae* type b, pneumococcus, and rotavirus infections), and 20 samples 1–77 days (median 53 days) post boost. Plasma was collected and stored at −70 °C until analysis. Plasma samples were heat inactivated at 56 °C for 30 min and clarified by centrifugation at 2400× *g* for 10 min. Supernatants were diluted 1:100 and analyzed using a commercial Tetanus Toxoid IgG ELISA kit (IBL America, Minneapolis, MN, USA) and Human Anti-Tetanus Toxoid IgM ELISA Kit (Alpha Diagnostic International, San Antonio, TX, USA) according to manufacturers’ guide.

### 2.5. Statistical Analysis

R v3.6.1 [35] was used for statistical analysis. The R packages phyloseq [37], vegan [38], and limma [39,40] were used to filter taxa with <5% prevalence and samples with fewer than 5000 read counts, to calculate distance matrices, and to perform differential abundance analysis, respectively. Alpha diversity was calculated without rarefication as suggested by McMurdie and Holmes [41]. Birth samples were excluded from differential abundance analysis, as 14 out of 24 birth samples had low read counts (<5000) due to a low DNA amount. Low-diversity samples (one taxon >90%; two postnatal samples) were further filtered prior to differential abundance analysis, as the normalized count would be highly skewed by the single dominating taxon. One of these low-diversity samples was dominated by *Coprobacillus* and one by *Bifidobacterium*; the latter sample was assessed by Bif-TRFLP. Supplementation at 1 month is defined as any formula supplementation in addition to breastfeeding by the age of one month. Unsupplemented infants were exclusively breastfed throughout the first month of life. To identify changed taxa associated with supplementation at 1 month, we performed limma–voom regression of bacterial abundance against the supplementation group while accounting for age, delivery method, the interaction between age and supplementation, and subject ID as a random effect. Bif-TRFLP results were compared using the Wilcoxon rank-sum test between supplemented infants and unsupplemented infants. The difference in *B. longum* representation between supplemented and unsupplemented infants is significant at one month (*p* = 0.042 by Wilcoxon rank-sum test, 0.026 by linear regression) and when considering data from both one week and one month (*p* = 0.030 by Wilcoxon rank-sum test, 0.017 by linear regression). When a variable for delivery method is included in the linear regression models, the corresponding *p* values are 0.055 and 0.045. Immune data were analyzed using a linear mixed effect model with the lmer function, accounting for age, supplementation, delivery method, and subject ID as a random effect. *p* values were adjusted for multiple testing using the qvalue package [42]. Results with adjusted *p* < 0.05 were considered significant. ggplot2 [43] was used to plot all figures.

## 3. Results

### 3.1. Different Gut Microbial Communities Associated with Different Ages, Diets, and Delivery Methods

In total, 105 samples were successfully sequenced at an average of 87,000 (10,491–1,537,440) reads per sample. Bacterial sequence counts were agglomerated to the genus level and reads not classified to the genus level were assigned to the lowest taxonomic assignment, resulting in 131 bacterial taxa. Taxa that were not present in at least 5% of the samples and samples with fewer than 5000 read counts were filtered, resulting in 90 samples and 67 taxa to be analyzed. Firmicutes, Proteobacteria, and Actinobacteria were the most abundant phyla found in all samples, accounting for 35%, 32%, and 22% of the total reads, respectively (Figure 1A). Approximately equal sample sizes were available for infants receiving or not receiving supplementation by one month of age (*n* = 11 and *n* = 13, respectively). Alpha diversities were calculated using Shannon diversity for each time point for infants supplemented with formula or exclusively breastfed, and no significant differences were observed (Figure 1B; Wilcoxon rank-sum test, *p* > 0.05). As expected, the diversity of microbial communities increased with age (Figure 1B). Principal coordinate analysis (PCoA) with weighted UniFrac distances demonstrated broad overlap between supplemented and unsupplemented infants in the first two axes (Figure 1C), but a significant difference in microbial communities of the two groups was detected by PERMANOVA (*p* = 0.026). Analysis of difference between male and female infants by PCoA showed broad overlap between the two groups (data not shown, PERMANOVA *p* = 0.045). PCoA also showed significant separation of samples by time point or delivery method (Figure 1D,E; PERMANOVA *p* < 0.05).

### 3.2. Delivery Method Influences the Infant Gut Microbiota through 6 Months of Age

Vaginally delivered infants (*n* = 16) had higher bacterial loads at birth compared to cesarean-delivered infants (*n* = 8), as 90% of the birth samples with >5000 read counts were from vaginal birth. We used the limma–voom pipeline to analyze and detect later differences (1 week, 1 month, 3 months, and 6 months) in the microbial community due to delivery method, treating time as a continuous fixed covariate. Eight taxa were differently abundant in the vaginally delivered vs. cesarean-delivered infants (Figure 2A; *p* < 0.05). Specifically, bacterial genera *Bifidobacterium*, *Bacteroides*, and *Parabacteroides* had significantly higher relative abundance in vaginally delivered infants (Figure 2A,B; adjusted *p* < 0.05). Analysis at each time point showed that these differences persisted continuously through six months of age (Figure 2C).

### 3.3. Changes in Microbial Communities Associated with Supplementation by 1 Month of Age

Analysis of all data using the limma–voom pipeline, treating time as a continuous fixed covariate while accounting for delivery methods, showed 18 taxa to be different in the microbiotas of supplemented vs. unsupplemented infants (Figure 3A; *p* < 0.05). Specifically, Bacteroidales family *S24-7* and the genera *Campylobacter*, *Dermabacter*, *Peptoniphilus*, and *Prevotella* were significantly higher in relative abundance in exclusively breastfed infants after allowing for false discovery; *Eggerthella* had a significantly higher relative abundance in supplemented infants (Figure 3A,B; adjusted *p* < 0.05). Analysis of differences at individual time points found that most of the significant differences were found at one month or three months, but that the differences between diets diminished by six months (Figure 3C). No significant differences were seen at one week after allowing for false discovery, showing that the microbiotas of supplemented and unsupplemented groups were similar prior to introduction of supplements.

Due to the known importance of *Bifidobacterium* in the infant gut environment, we further used Bif-TRFLP to assess *Bifidobacterium* species in these samples. Subsequently, 75 samples containing taxa identified to the family level of *Bifidobacteriaceae* in the metagenomics experiment were investigated, which revealed 59 samples with identifiable *Bifidobacterium* species (Figure 3D). *B. longum* was found to be statistically different at one month between supplemented and unsupplemented infants (Figure 3E; Wilcoxon rank-sum test, *p* < 0.05). When delivery method is included as a covariate, representation of *B. longum* remains different (*p* = 0.045) considering aggregated one-week and one-month data, and nearly so (*p* = 0.055) when considering one-month data alone. However, by 6 months of age there was no association between abundance of *B. longum* and having received supplementation by 1 month of age.

### 3.4. Characterization of Immune Development and Vaccine Responses of Supplemented and Unsupplemented Infants

We analyzed 12 cell-surface markers by flow cytometry and collected information about 23 interpretable immune phenotypes representing combinations of these markers (Appendix A). When plotted by the first and second principal components, immune phenotypes did not change with supplementation (Figure 4A) or delivery methods (PERMANOVA, *p* > 0.05), but significantly clustered by time point (Figure 4B; PERMANOVA, *p* < 0.05). We did not find any significant differences in immune-cell subsets of supplemented and unsupplemented groups at any time point (Wilcoxon rank-sum tests, *p* > 0.05). We also used linear mixed-effect models accounting for supplementation, delivery method, age, and subject ID as a random effect to determine if longitudinal development of immunophenotypes correlated with supplementation or delivery method; however, no significant interaction was observed (*p* > 0.05). As expected, however, developmental trajectories of many immune-cell subsets were seen to correlate with the age of infants (Figure 4C; adjusted *p* < 0.05). Additionally, anti-tetanus toxoid IgG levels were not significantly different between supplemented or unsupplemented infants (Figure 4D, *p* > 0.05). Anti-tetanus toxoid IgG did not correlate with abundance of bacteria in the genus *Bifidobacterium* (Spearman rank correlation coefficient = 0.086, *p* > 0.05), or with the abundance of individual species, at 3 months and 6 months of age. High concentrations of anti-tetanus toxoid IgG were seen in the immediate postnatal period and then declined throughout the study period, suggesting that maternal IgG masked antibodies produced *de novo* by the infants after receiving the tetanus vaccine at 2 and 4 months. Anti-tetanus toxoid IgM levels increased after vaccination, but no significant differences were observed between supplemented and unsupplemented infants (Figure 4E; *p* > 0.05).

## 4. Discussion

Our results show that early supplementation with formula to support infant nutrition when breast milk supply is inadequate has no major or persistent effects on microbiome-mediated infant health before the age of six months. The importance of the human microbiome, the “forgotten organ” [44], has been established in health and disease states [9,12,13,14,15]. The initial colonization of the gut in early infancy may be especially important as it has been suggested to have long-term health effects [45]. While some effects of exclusive breastfeeding or exclusive formula feeding are clear, studies examining the impact of combining breastfeeding with supplementation have reported disparate results [46]. A recent study suggested that geographical and socioeconomic differences may contribute to differences in health outcomes due to breastfeeding [24]; thus, previous studies in low-income settings might not apply to higher-income settings. There is very little information on the effects of mixed-feeding regimens in subjects in developed countries, and no studies documenting the long-term effect of supplementation delivered in the first month of life. In this study, we found that supplementation caused transient changes in the infant microbiota but no long-term effects. In contrast, the effect of delivery method on infant gut microbial composition has been shown in this and other studies [47,48,49,50,51,52] to persist through the age of six months. Immunophenotypic development did not correlate with changes in microbiota that resulted from differences in supplementation.

We found that bacterial genera *Bifidobacterium*, *Bacteroides*, and *Parabacteroides* were significantly more abundant in vaginally delivered infants. Changes in the infant gut microbiota due to differences in delivery method are well documented [47,48,49,50,51,52] and were demonstrated again in our results. Previous epidemiological studies have shown an association between cesarean delivery and poor health outcomes, including childhood asthma [53,54] and atopy [55]; however, the exact mechanism is unknown. Our study showed that differences in microbiota associated with delivery can be observed as late as six months of age, but these differences were not correlated with any changes in immunophenotypes or tetanus-vaccine responses, suggesting that more detailed studies of immune function might be important in future studies.

Exclusively breastfed infants had significantly higher levels of *S24-7*, *Campylobacter*, *Dermabacter*, *Peptoniphilus*, and *Prevotella*, but a lower level of *Eggerthella*, compared to supplemented infants. The significant differences observed between supplemented and unsupplemented infants are in agreement with prior studies looking at changes in microbiotas due to dietary differences. Specifically, the increase in Bacteroidales family *S24-7* was seen to associate with the supplementation of milk fat globule membrane in mice, which is an important component lacking in formula [56]. On the other hand, *Peptoniphilus* was commonly found in human breast milk and correlated with the content of lactose and non-fatty solid contents in the breast milk sample [57]. *Prevotella* was found to be more abundant in the oral microbiome of breastfed infants [58] and in the stool of breastfed *Rhesus macaque* [59]. The increased abundance of *Eggerthella* in supplemented infants is likely due to the absence of the genus during early timepoints (<1 month old), which agreed with a prior study where researchers found that *Eggerthella* colonization happened after 8 weeks in breastfed infants [60]. *Dermabacter*, a skin-associated microbe [61], has higher abundance in exclusively breastfed infants possibly due to an increase in direct skin contact with the mother. The increased abundance of *Campylobacter* in breastfed infants is of potential concern because *Campylobacter* is a pathogen that can cause diarrhea. Previously, breast milk was thought to protect infants against *Campylobacter* infection due to the presence of antibodies and human milk oligosaccharides in breast milk [62,63]. However, in sub-Saharan African and South Asian infants, *Campylobacter* abundance was shown to be correlated with breastfeeding, and resulted in more diarrhea cases in breastfed infants [64]. The increase of *Campylobacter* was also observed in breastfed *Rhesus macaque* compared to that in bottle-fed animals, which correlated with the increase in arachidonic acid present in the stool and was hypothesized to contribute to T_H_17 cell expansion [59]. Further investigation into the species of *Campylobacter*, their metabolism capability, and the nutrient contents of the breast milk are needed to determine the role of breast milk and *Campylobacter* infection. Previous studies of infant microbiomes have been largely focused on *Bifidobacterium* because of this taxon’s unique ability to ferment human milk oligosaccharides and affect long-term health [10,11]. While we did not find any differences in *Bifidobacterium* abundance by 16S rRNA gene sequencing alone, further typing of *Bifidobacterium* species using Bif-TRFLP revealed that *Bifidobacterium longum* is significantly less abundant during early infancy for infants supplemented at 1 month of age; however, importantly, the difference observed during early infancy dissipated by six months.

While a transient change in microbiota with supplementation is expected, it is noteworthy that these differences were not associated with any measurable differences in immune development. Specifically, breastfeeding with supplementation did not result in reduced T-cell activation (HLA-DR^+^) as previously observed in exclusively formula-fed infants [65]. It is proposed that infant diet is linked to health outcomes via changes in the microbiome and associated downstream changes in host metabolism; for example, clinical studies with infants in Bangladesh showed that a higher abundance of *Bifidobacterium* was correlated with enhanced thymic development and better vaccine responses [13,30]. However, in the United States where women were offered tetanus–diphtheria–pertussis (Tdap) vaccination during pregnancy, maternal IgG transfer resulted in no correlation between neonatal IgG and *Bifidobacterium* levels [66,67]. Additionally, the abilities of infants to secrete IgM against tetanus toxoid after vaccination were similar. One caveat of microbial 16S rRNA gene sequencing is that only the identities of microbes are detected, not their functional capacities. The ability of *Bifidobacterium* species to ferment human milk oligosaccharides has been shown to be strain-specific and dependent on the geographical locations of subjects [24,68]. Detection of microbial functions through metabolomics or meta-transcriptomics could reveal important links between the microbiota and host immunity.

We longitudinally assessed both microbiota composition and immunophenotypes in parallel and were able to establish correlations with breastfeeding habits. The main limitations of our study are small sample size, the small amount of blood we were able to collect from infants, and sporadic data on type of formula used for supplementation. A larger sample size would allow for clearer inference of any effect of delivery method on immunophenotypes that is mediated by microbiota composition. The sample obtained here was sufficient to allow monitoring of 23 important immune-cell populations, including a variety of activated cells that should reflect the overall level of antigenic stimulation causing activation and memory-cell differentiation. However, future studies with larger blood samples would allow for more functional testing of these immune cells, including adaptive T-cell responses to vaccination, which is another important measure of vaccine response that is independent of maternal influence. Because this is a prospective study and focused on effects of any supplementation, the amount and types of formula received by the infants may differ and thus limit our ability to detect differences between groups. Nevertheless, our data show that supplementation of infants by one month of age was associated with no detectable change in the major circulating immunophenotypes.

## 5. Conclusions

Milk, intestinal microbiota, and immunity (MIMI) is a prospective, longitudinal study of how infant dietary intake affects the intestinal microbiota and immune development. We found that differences in infant microbiotas were largely determined by delivery methods and not formula supplementation, consistent with previous studies [51]. Supplementation caused transient changes in the microbiota, notably increases in *Campylobacter*, *Dermabacter*, *Peptoniphilus*, *Prevotella*, and *S24-7*; and a decrease in *Eggerthella*. However, these differences diminished by the age of 6 months. No differences were detected in the immune development of supplemented vs. unsupplemented infants, whether in immunophenotypes or responses to the tetanus vaccine. Thus, early supplementation to support infant nutrition when breast milk supply is inadequate had no detectable detrimental effects on microbiome diversity or immune function.

## Figures and Tables

**Figure 1 nutrients-13-00807-f001:**
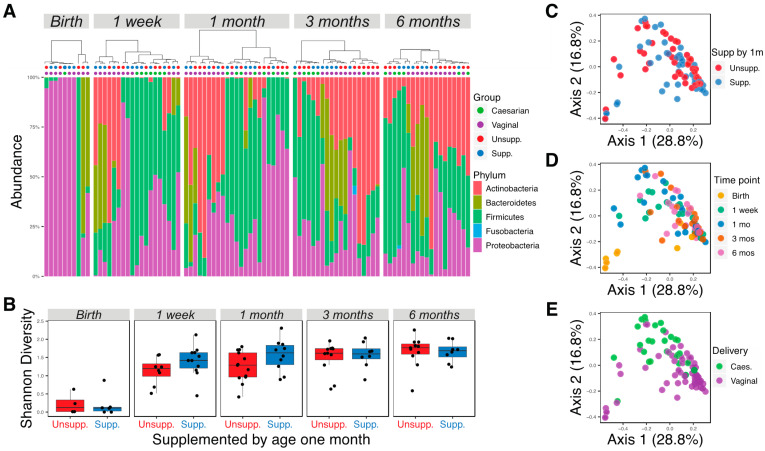
Microbial community diversity increased with age and differed according to supplementation and delivery method. (**A**) Relative abundance of bacteria at the phylum level of all infants at different time points. Samples were ordered using weighted UniFrac distance and Ward D clustering method. Colored dots topping each column indicate whether the infant was delivered through cesarean or vaginally; and if unsupplemented or unsupplemented by one month of age. (**B**) Shannon diversity of infant microbiotas at different time points. (**C**) PCoA plot generated using weighted UniFrac distance, colored by diet difference at the age of one month. (**D**) PCoA plot generated using weighted UniFrac distance, colored by time point. (**E**) PCoA plot generated using weighted UniFrac distance, colored by delivery method. Unsupp.: unsupplemented; Supp.: supplemented.

**Figure 2 nutrients-13-00807-f002:**
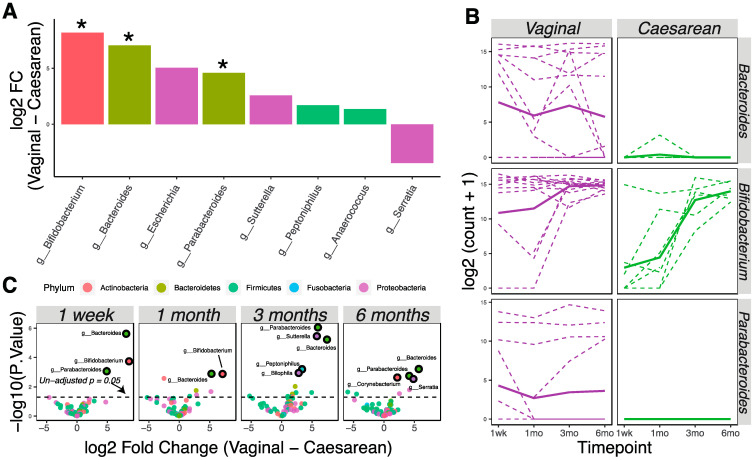
Delivery methods affect the infant microbiota through six months of age. (**A**) Bacterial taxa that were differentially abundant as detected by the limma–voom regression method (*p* < 0.05; * adjusted *p* < 0.05). Taxa with positive numbers are more abundant in vaginally delivered infants; taxa with negative numbers are more abundant in cesarean-delivered infants. Taxa were colored according to phylum membership. (**B**) Representation of significantly changed taxa at each time point in vaginally delivered vs. cesarean-delivered infants. Solid lines show the mean abundance; dashed lines show the individual level. (**C**) Volcano plots show significantly different taxa between vaginally delivered and cesarean-delivered infants at different time points. Markers for taxa having significant adjusted *p* values (<0.05) are circled in black.

**Figure 3 nutrients-13-00807-f003:**
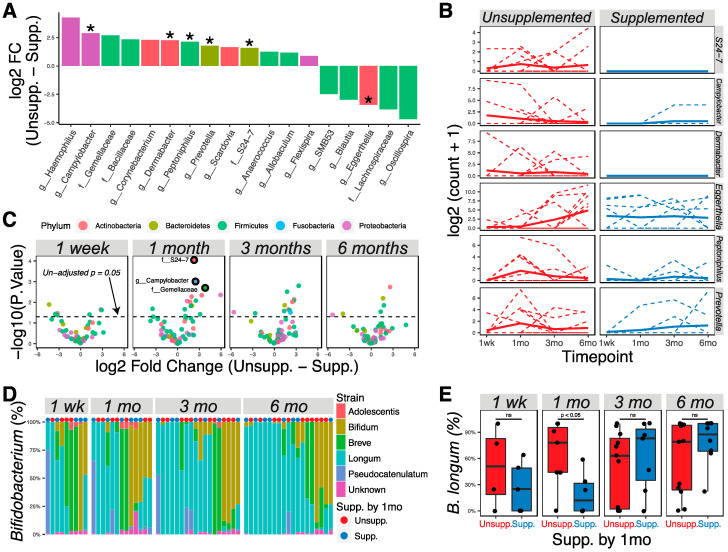
Supplementation by 1 month of age was correlated with abundance of specific microbial taxa at 1 month and 3 months of age. (**A**) Bacterial taxa that were differentially abundant as detected by the limma–voom regression method (*p* < 0.05; * adjusted *p* < 0.05). Taxa with positive numbers are more abundant in unsupplemented infants; taxa with negative numbers are more abundant in supplemented infants. Taxa were colored according to phylum membership. (**B**) Representation of significantly changed taxa at each time point in unsupplemented and supplemented infants. Solid lines show the mean abundance; dashed lines show the individual level. (**C**) Volcano plots show significantly different taxa between unsupplemented and supplemented infants at different time points. Markers for taxa having significant adjusted *p* values (<0.05) are circled in black. (**D**) Bif-TRFLP analysis of *Bifidobacterium* species. Samples were ordered using the Bray–Curtis dissimilarity index and Ward D clustering method. (**E**) Boxplots showing the percentage abundance of *Bifidobacterium longum* among all bifidobacteria at different time points.

**Figure 4 nutrients-13-00807-f004:**
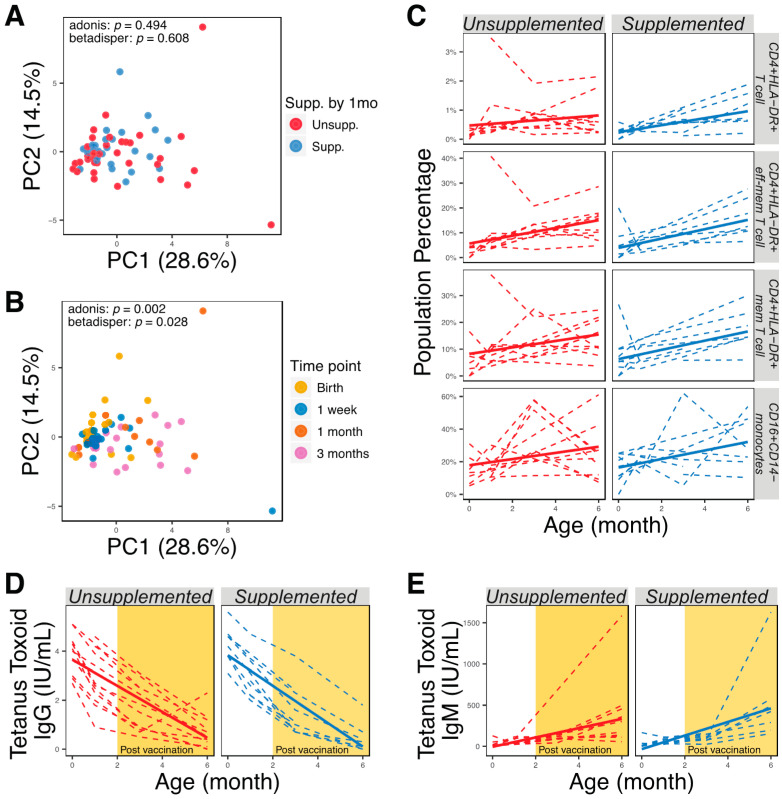
Similar immune development of supplemented and unsupplemented infants. (**A**) PCA plot generated using all immune markers measured, colored according to supplementation by 1 month. (**B**) PCA plot generated using all immune markers measured, colored by time point. (**C**) Immune cell populations that were significantly correlated with age (adjusted *p* < 0.05). (**D**) Quantification of anti-tetanus toxoid IgG in serum using ELISA. (**E**) Quantification of anti-tetanus toxoid IgM in serum using ELISA.

**Table 1 nutrients-13-00807-t001:** Characteristics of enrolled participants.

Variable	*n* = 24
Mother primiparous	15 (63%)
Race	
White non-Hispanic	17 (71%)
Hispanic	1 (4%)
Southeast Asian	3 (13%)
West Asian	3 (13%)
Delivery method	
Vaginal	16 (67%)
Cesarean	8 (33%)
Sex	
Female	10 (42%)
Male	14 (58%)
Diet	
Any breastfeeding at 1 week	24 (100%)
Breastfeeding without formula at 1 week	19 (79%)
Any breastfeeding at 1 month	24 (100%)
Breastfeeding without formula 1 month	13 (54%)
Any breastfeeding at 3 months	23 (96%)
Breastfeeding without formula at 3 months	13 (54%)
Any breastfeeding at 6 months	22 (92%)
Breastfeeding without formula 6 months	3 (21%)
Semi-solid or solid foods at 6 months	17 (81% of 21; information missing for 3)

## Data Availability

The datasets generated and analyzed during the current study are available in the SRA under BioProject ID PRJNA702525 and GitHub (https://github.com/HOC-Lab/mimi/, accessed date 20 March 2020).

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
