# Peer review of "Transient Effect of Infant Formula Supplementation on the Intestinal Microbiota"

_nutrients, 2021, doi:10.3390/nu13030807_

Round 1

Reviewer 1 Report

Chin et al. investigated the transient effect of infant formula supplementation on the intestinal microbiota of initial breast milk fed infants. The authors also attempted to measure the immune cells and immune response after vaccination.  This is an important issue since most infants in the United States are not exclusively breastfed in the first six months. Data or samples were collected at 5 time points (at birth; 1 week; and 1, 3, and 6 months of age). The time points were good for trajectory study. However, the enrollment of infants could be difficult and the exposed amount of formula were not defined. The paper could be improved after some definitions were clarified.

About the feces at birth, is it meconium or transition stool? What is the timing? day 0 or day 3 after birth? Meconium may contain much less microbiota than the transition stool.  It is an important information to be mentioned in the methods.

Do the author review any antibiotics exposure history in this cohort? is it included in the data on receipt of immunization? It is an important information to be mentioned in the results or limitation section. How about the sex proportion? Sex has been reported as an important variable for gut microbiota.

In this small size cohort, the authors seemed to use two equal size subgroups— unsupp. and  supp. breast milk fed infants. Figure 3E showed significant difference of percentage in B. longum. The mode of delivery may be an important confounder of this result. Is the proportion of vaginally delivered infants equal in two sub-groups? The result of Wilcoxon rank sum test should be carefully explained and discussed.

Following the above question, total 19 subjects provide the stool sample at 1 months and 6 months in line 107–110. How were the followed-up 19 subjects located in each subgroups?  The authors should report the follow-up in each group.

The reviewer could not find the final number of blood samples. It is an important information to be mentioned in the results.

In line 123, what is the innate cells? Could the author provide the abundance results of immune cells for each time point as the supplementary result? It could provide more value for this paper.

The authors should be cautious of interpretation of the IgG level. The concentration was high than the reported study (PMID: 23585389). Did those mothers receive antenatal vaccine? Infants born to tetanus-toxoid immunized mothers were reported to have lower tetanus antitoxin titres after vaccination than infants whose mother are not immunized.

The reviewer agreed to use IgM as a marker to explain the immunity response. It is a smart approach for measure immunity response. However, IgM level after tetanus toxoid exposure changed in a small time windows and depends on primary and secondary vaccination. The author should provide the detailed time of capillary blood sampling after vaccination in the result.

Author Response

Dear Editors,

We thank the reviewers for their insightful, constructive comments.  We have revised the manuscript and believe it to be substantially improved.

Revisions in the manuscript are indicated using Word’s Tracked Changes feature.

Comments and Suggestions for Authors

Point 1: About the feces at birth, is it meconium or transition stool? What is the timing? day 0 or day 3 after birth?....It is an important information to be mentioned in the methods.

  • The feces at birth were meconium collected at day 0. This information has been added in line 97 of Materials and Methods.

Point 2: Do the author review any antibiotics exposure history in this cohort? is it included in the data on receipt of immunization?....How about the sex proportion? Sex has been reported as an important variable for gut microbiota.

  • We only recruited infants who had no antibiotic exposure postnatally. Details of antibiotic exposure and the sex proportions are now indicated in line 90 in Materials and Methods and Table 1, along with comment about the results of an analysis of sex in line 197-198 in Results.

Point 3: In this small size cohort, the authors seemed to use two equal size subgroups— unsupp. and  supp. breast milk fed infants. Figure 3E showed significant difference of percentage in B. longum. The mode of delivery may be an important confounder of this result. Is the proportion of vaginally delivered infants equal in two sub-groups? The result of Wilcoxon rank sum test should be carefully explained and discussed.

  • The difference in B. longum representation between supplemented and unsupplemented infants is significant at one month (p=0.042 by Wilcoxon test, 0.03 by linear regression) and also when considering data from both one week and one month (p=0.03, 0.02). When a variable for delivery method is included in the linear regression models (to account for its effect) the corresponding p values are 0.06 and 0.04.  This information is now included in the Materials and Methods, lines 168-174, and in Results, line 248-250.

Point 4: …19 subjects provide the stool sample at 1 months and 6 months in line 107–110. How were the followed-up 19 subjects located in each subgroups?

  • This information is now provided in Materials and Method, lines 112-116.

Point 5: The reviewer could not find the final number of blood samples.

  • Information on blood samples was added to Materials and Method, line 129.

Point 6: In line 123, what is the innate cells? Could the author provide the abundance results of immune cells for each time point as the supplementary result?

  • We have specified the innate cells surveyed in Materials and Method line 129. We have added Supplemental Table S1 showing the average representation of immune cells at each time point, in supplemented and unsupplemented groups. The Supplemental Table is referenced in Results line 268 and Supplementary Materials line 408-410.

Point 7: The authors should be cautious of interpretation of the IgG level. The concentration was high than the reported study (PMID: 23585389). Did those mothers receive antenatal vaccine?

  • We did not specifically query mothers about antenatal vaccine, but it is the standard practice, as referenced in ref no. 31. This information is now included in Materials and Methods, line 90-92. The range of anti-tetanus IgG concentration in our cohort at birth is 2.7 – 5.6 IU/mL (median = 2.45 IU/mL), which is within the range of anti-tetanus IgG detected at birth to HIV-unexposed infants (Fig. 2 bottom left, PMID: 23585389).

Point 8: The author should provide the detailed time of capillary blood sampling after vaccination in the result.

  • We have added details relevant to the time frame of blood collection after prime and boost in Materials and Method, line 142-146.

Reviewer 2 Report

This is the well-prepared manuscript, but I have a few comments:

I didn't find information about infant formula content; was the same formula for all infants? Please clarify.

In my opinion, the discussion should be rewritten. At the beginning, describe your findings in the first paragraph. In the next paragraphs, you first give your result and then discuss it.

There are no limitations (there is only one) and the strengths of your research and most importantly, there are no conclusions.

Author Response

Dear Editors,

We thank the reviewers for their insightful, constructive comments.  We have revised the manuscript and believe it to be substantially improved.

Revisions in the manuscript are indicated using Word’s Tracked Changes feature.

Point 1: I didn't find information about infant formula content; was the same formula for all infants?

  • Because this is a prospective study focused on if the infants were supplemented at all, the amount and types of formula received by the infants differ. We only have sporadic data. We added a discussion of this limitation (lines 391-393).

Point 2: At the beginning [of the discussion], describe your findings in the first paragraph. In the next paragraphs, you first give your result and then discuss it.

  • We now describe our findings in the first sentence of the first paragraph and have reiterated relevant results in the first sentence of each of the following paragraphs (lines 296-298, 317-318, and 329-331).

Point 3: There are no limitations (there is only one) and the strengths of your research and most importantly, there are no conclusions.

  • We have added details describing our limitations and strength (line 379-385). We have also added a conclusion section (line 396-407).

Round 2

Reviewer 1 Report

The reviewer agreed with the authors' response to previous mentioned questions. 
Only one small question is the unit of the Supplementary Table S1. Is it percentage? And % to which cells? Please clarify it in the table footer. 

Author Response

Dear Editors,

Thank you for the reminder. The appropriate units specifying the % of different cell types are now added to Supplementary Table 1.